# Long-Term Forecasting Potential of Photo-Voltaic Electricity Generation and Demand Using R

**Karina Vink [1,2,3,\*], Eriko Ankyu [1,4] and Yasunori Kikuchi [1,5]**

[1] Technology Integration Unit, Global Research Center for Environment and Energy based on Nanomaterials Science (GREEN), National Institute for Materials Science (NIMS), Tsukuba, Ibaraki 305-0044, Japan (during study); ankyu.eriko.gn@u.tsukuba.ac.jp (E.A.); ykikuchi@ifi.u-tokyo.ac.jp (Y.K.)

[2] Department of Water Engineering and Management (WEM) Faculty of Engineering Technology, University Twente, 7522 NB Enschede, the Netherlands

[3] Department of Construction Management and Engineering (CME), Faculty of Engineering Technology, University Twente, 7522 NB Enschede, the Netherlands

[4] Faculty of Human Sciences, University of Tsukuba, Tsukuba, Ibaraki 305-8577, Japan

[5] Institute for Future Initiatives, the University of Tokyo, Tokyo 113-8654, Japan

\* Correspondence: k.vink@utwente.nl; Tel.: +31-53-489-2547

**Abstract:** For micro-grid cost-benefit analyses, both energy production and demand must be estimated on the long-term of one year. However, there remains a scarcity of studies predicting energy production and demand simultaneously and in the long-term. By means of programming in R and applying linear, non-linear, and support vector regression, we show the in depth analysis of the data of a micro-grid on solar power generation and building energy demand and its potential to be modeled simultaneously on the term of one year, in relation to electricity costs. We found solar power generation is linearly related to solar irradiance, but the effect of temperature on total output was less pronounced than anticipated. Building energy demand was found to be related to multiple parameters of both time and weather, and could be estimated through a quadratic function in relation to temperature. Models for both solar power generation and building energy demand could predict electricity costs within 8% of actual costs, which is not yet the ideal accuracy, but shows potential for future studies. These results provide important statistics for future studies where building energy consumption of any building type is correlated in detail to various time and weather parameters.

**Keywords:** energy demand; long-term forecasting; machine learning; R programming; solar power generation; support vector regression

---

## 1. Introduction

Increasing the share of renewable energy is crucial to overcome climate change, and to target 7.2 of the sustainable development goals to be reached by 2030. Micro-grids are increasingly becoming more popular as solutions for helping to reach renewable energy goals [1]. Micro-grids allow for local energy production, often combined with local energy storage, and often combined with an awareness or ambition to reduce overall energy demand. As micro-grids involve the management of both energy production and demand, both of these sides need to be predicted simultaneously in order to generate future cost benefit analyses. What we have found is that there is still a wide gap between predicting energy production and demand, in terms of scale, time, methodology, and software.

In this study we aim to determine the potential of long-term forecasting (at least one year) of energy production and demand on the scale of a research building by means of programming with the software R and applying linear, non-linear, and support vector regression. The research building

in question is the National Institute of Materials Science (NIMS) NanoGREEN/WPI-MANA building in Tsukuba, Japan, which has a micro-grid consisting of four arrays of solar panels and has been in operation since 2013. The raw data per second of this micro-grid contained gaps and erroneous data, which have been cleansed in a previous study as described [2]. Using the improved data records from 2015 to 2017, we evaluated the potential of predicting future energy demand and energy generation.

Published studies on optimizing building energy consumption have increased 10 fold in the past 10 years [3]. This has included numerous variations of machine learning studies, which have explored either the forecasting of building energy demand [4], or renewable energy generation and weather patterns (for solar irradiance) [5], separate from each other. However, for micro-grid costs estimations it is necessary to consider these two factors simultaneously. A second shortcoming of previous studies is that they tend to focus on short term predictions and neglect long-term predictions. For micro-grid cost and benefit analyses we require long-term future predictions of the amount of solar power generated and the energy demand of the building. What is considered long- and short-term is defined differently for energy demand and weather prediction, as seen in Appendix A. According to the sources in Appendix A, the long-term for solar irradiance, our main weather parameter of interest, ranges from one day to two months, whereas for energy demand this is 1-10 years. Ideally it would be useful to predict the costs and benefits of the micro-grid one year ahead, with monthly averages and hourly estimates, as the electricity price varies per hour. We expect to see reduced solar power generation over time as the efficiency of the solar panels decreases, and an increase in building energy demand due to higher occupancy and laboratory use over the years. So far no study has considered how to incorporate both energy demand and solar power generation by means of machine learning for the long-term of one year. To reach this, it is important to consider not just the optimum results from higher-level regression analysis, but also the underlying physical and chemical mechanisms causing the found results in the actual data and suggest practical solutions based on these principles.

The hypotheses for solar power generation are the following. Larson et al. [6] showed a linear relationship between solar power generation and solar irradiance. However, it is also shown by Dupré et al. [7] that temperatures higher than 25 degrees Celsius negatively affect the capability of crystalline silicon to generate electricity. Similarly, it should perform between 1 and 1.1 times the normalized efficiency between 25 and 0 degrees Celsius, and presumably even high at colder temperatures. It is therefore expected to see an influence of temperature as well. Degradation is expected to lead to reduced efficiency in 2017 as compared to 2015.

The hypotheses for building energy demand are the following: the total building energy demand consists of purchased electricity, solar power generation, and battery discharging. The amount of generated solar power is at most around 10% of the total building energy demand and is immediately consumed by the building [8]. To accurately link back to electricity prices, only purchased electricity is considered as it is the main contributor. It is expected that building energy demand depends on building occupancy as well as weather circumstances, with colder and warmer temperatures leading to higher use of air-conditioning.

## 2. Materials and Methods

### 2.1. Data Description

An overview of the parameter names and units is given in Table 1.

- Micro-grid data, two parameters. An initial cost analysis was performed on the hourly data from the micro-grid by [8] from April 2013 to March 2018. These data were recorded and obtained from a different data storage point in the Energy Management System and not yet cleansed. The data used in this study concerns per second data, from January 2015 to December 2017, which was obtained from the micro-grid data storage point and was cleansed as described in [9]. It was converted into hourly data to enable analysis with hourly weather data by means of averaging. Parameters of interest are purchased electricity and total solar power generation.

For the purpose of this study, battery charging and discharging is ignored as it is of negligible proportions. The electricity amount and cost reduction function of the battery is active only during July, August, and September. Previous studies showed this amount is maximally 4.86% of the total building energy demand [10].

- Japan Meteorology Agency (JMA) data, six parameters. The hourly weather parameters for the corresponding time period of January 2015 to December 2017 were downloaded in csv files from the JMA website (freely accessible) [11] in batches of one month due to size restrictions. Parameters of interest are temperature, precipitation, sunlight, wind speed, solar irradiation, and humidity.
- Time parameters (6): year (note that due to the total data volume each year is calculated separately), month, day of month, day of week, hour, holiday.
- Electricity cost data, one parameter (see Figure 1). These data were obtained from the Facility Management Office of NIMS, based on the yearly renewed contract with the Tokyo Electric Power Company. The Feed in Tariff surcharge for renewable energy is included. Other factors such as fuel regulatory costs, basic connection to the electricity grid and spare line fees are not included since these are shared by the entire site and independent of building electricity use.

**Table 1.** Parameter names and units.

| Data Source | Parameter Name | Parameter Unit |
|---|---|---|
| NanoGREEN/WPI-MANA micro-grid | total_pv_ac<br>purchased_electricity_kwh | kWh (alternating current)<br>kWh |
| Japan Meteorological Agency (JMA) | temperature<br>precipitation<br>sunlight<br>wind_speed<br>solar_irradiation<br>humidity | °C<br>mm<br>numeral (between 0–1 h)<br>m/s<br>W/m$^2$<br>% |
| Time parameters | month<br>day_of_month<br>hour_end<br>day_of_week<br>holiday | month (1–12)<br>day (1–31)<br>hour (1–24)<br>day (1–7, starting on Monday)<br>numeral (0 for no holiday, 1 for holiday) |
| National Institute of Materials Science (NIMS) Facility Management Office | electricity price | JPY (Japanese Yen) |

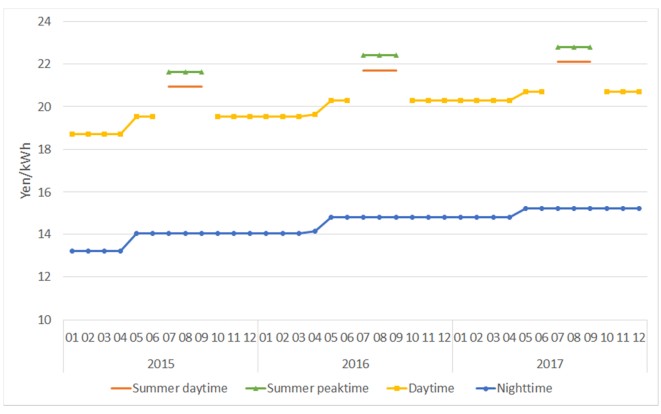

**Figure 1.** Electricity costs per time period (Japanese Yen/kWh).

## 2.2. Approach

Machine learning is the ability of a computer to learn from data without explicit programming. The machine learns from the existing data and predicts future data. In supervised learning, algorithms build a mathematical model of a set of data that contains both the inputs and the desired outputs,

so their classification is known [12]. Supervised machine learning can be applied to the data in two forms:

- Predictive modeling to determine what would be the output if a new (future) input is applied (to estimate future demand and solar power generation)
- Prescriptive modeling to optimization what actions should be taken given the data (to estimate the impact of additional storage devices)

In this study we want to examine the potential of predicting two parameters: solar power generation and building energy use. We endeavor to evaluate the prediction potential of both parameters with one model using the same input parameters for weather and time, to see to which extent accurate electricity cost forecasts can be achieved.

### 2.3. Motivation for Software Choice

Various software is available for performing machine learning. Given the overview of advantages and disadvantages shown in Appendix B, we chose the software R (R version 4.0.1) mainly as it deals efficiently with large datasets, is freely available, and has high data analysis and visualization potential.

### 2.4. Data Cleansing

After the initial cleansing and averaging of the per second micro-grid data as per [9], additional cleansing steps were required. The JMA weather data similarly required cleansing. The cleansing of both datasets are described below. In all cases of missing data, the entire row of data was removed so as not to lead to invalid correlations.

1. Removing missing values from the micro-grid data and replacing negative values with 0. For the micro-grid, this concerned the maintenance days in November (2015: November 14, 8:00–20:00, 2016: November 19, 8:00–19:00, 2017: November 18, 8:00–18:00). Two additional instances were found to have missing values (2015/11/13 12:00, 2017/09/25 21:00). For solar power generation, 269 negative values were recorded during nighttime in 2017, which were replaced with 0.

2. Removing JMA data with poor data quality. The JMA has excellent record keeping concerning data quality. Most values have the data quality value of "8: There are no missing data on the statistical basis (normal value)". The number of instances when the data quality was less than this are described in Table 2. Values with data quality values of 4 (10 instances) or 5 (2 instances) were kept. There were several instances where data was missing for multiple parameters at the same time.

3. Removing outliers (see SM1: R code for outlier removal). Note that the parameter (relative) humidity is in % unit, and sunlight is relative to one full hour (1), and therefore these parameters should not have outliers removed. The hourly electricity price is given based on predetermined conditions and cannot have outliers. Precipitation occurs intermittently and in irregular amounts, meaning it is prone to outliers. This leaves three weather parameters and the two micro-grid parameters to remove outliers for. For the parameters "solar irradiation" and "total pv ac" outliers were calculated without taking the values of 0 into account. Extreme outliers were removed by calculating three times the interquartile range (iqr) using the following code:

$$\text{extreme\_threshold\_upper} = (\text{iqr} * 3) + \text{upperq} \tag{1}$$

$$\text{extreme\_threshold\_lower} = \text{lowerq} - (\text{iqr} * 3) \tag{2}$$

Boxplot results were used to show the difference before and after outlier removal. As an example, Figure 2 shows the data before and after removing outliers for the parameter wind speed, using data from 2015. Outliers were checked for overlap between parameters and consequently removed. Before outlier removal the standard deviation, mean, minimum and maximum values

were calculated for the five parameters: wind speed, temperature, solar irradiation, total pv ac, and purchased electricity, for each year. Table 3 gives an overview of the found outliers.

4. Removing additional outliers occurring around maintenance days. After removal of the values in steps 1–3, we observed several remaining uncharacteristic data values occurring around the maintenance days. As example, from November 13 to 16 in 2015 the values for purchased electricity from the micro-grid were uncharacteristically low. The cause of these low values is most likely the shutdown of electricity dependent equipment in anticipation of and in reaction to the maintenance procedures, during which all electricity through the main power line is stopped. This led to the removal of 47 rows of data in 2015; 47 in 2016 and 49 in 2017. After this final removal step the standard deviation, mean, minimum and maximum values were calculated for the five parameters wind speed, temperature, solar irradiation, total pv ac, and purchased electricity, for each year.

**Table 2.** Overview of data quality values of JMA parameters.

| # | JMA weather parameter data quality | 5: There is a deficit of 20% or less of the data that is the basis of statistics (quasi-normal value) | | | 4: There is a deficit exceeding 20% in the data underlying the statistics (lack of data) | | | 1: There is no statistical value (missing) | | |
|---|---|---|---|---|---|---|---|---|---|---|
| Year | | 2015 | 2016 | 2017 | 2015 | 2016 | 2017 | 2015 | 2016 | 2017 |
| 1 | Temperature | - | - | - | - | - | - | - | - | 2 |
| 2 | Precipitation | - | - | - | - | - | - | - | - | - |
| 3 | Sunlight | - | - | 1 | - | 1 | 3 | - | - | 22 |
| 4 | Wind speed | - | - | - | - | - | - | - | 1 | 2 |
| 5 | Solar irradiance | 1 | - | - | 2 | 2 | 2 | 1 | - | 2 |
| 6 | Humidity | - | - | - | - | - | - | - | - | 2 |

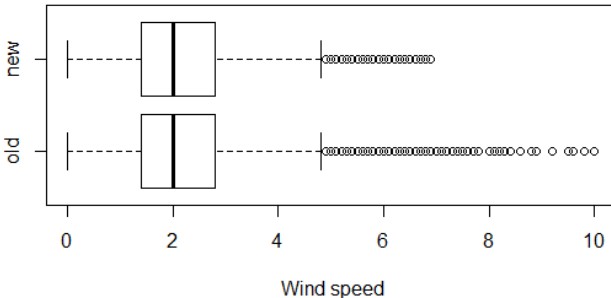

**Figure 2.** Data points before and after removing outliers of the parameter "Wind speed" in data from 2015.

**Table 3.** Outliers removed per parameter and per year.

| Parameter Name | 2015 | 2016 | 2017 |
|---|---|---|---|
| wind_speed | 88 | 68 | 100 |
| temperature | 0 | 0 | 0 |
| solar_irradiation | 0 | 0 | 0 |
| total_pv_ac | 0 | 0 | 0 |
| purchased_electricity_kwh | 2 | 0 | 0 |

*2.5. Data Exploration (See SM2: R Code for Data Exploration Graphs)*

The main exploration of the relationships between the different parameters was performed through means of plotting different combinations of parameters:

- Generated solar power/building energy demand in relation to various individual parameters per month and hour of the day, with values color coded by e.g., temperature, based on Boeye [13].
- All parameters correlated to each other through the ggpairs package, based on [14].

*2.6. Solar Power Generation–Linear Regression and Support Vector Regression (See SM3: R Code for Linear Regression and SVR (Support Vector Regression))*

To test the relationship of solar power generation (parameter name: total_pv_ac) and solar irradiation, the SVM package e1701 for R was applied as described in [15]. This procedure first calculated the root square mean error by means of a linear regression model, after which improvement was attempted through support vector regression, leading to a non-linear model. As is standard, the parameter epsilon (margin band width) should be higher than 0 to prevent overfitting, but not too high as then too few support vectors are selected. Similarly, the parameter c (cost) should be higher than 0 in order to penalize some points, but not too high which would have all points within the margin penalized. The calculated ideal epsilon and cost values were plotted in a graph. The initial models took just over 1 h to generate per year. Tuning the models took around 3 h for each year. The total calculation time was over 12 h. For those readers interested in designing similar methods in greater detail, we refer to Karatzoglou [16].

*2.7. Building Energy Demand–Non-Linear Regression (See SM4: R Code for Plotting Nonlinear Regression)*

Data exploration showed that purchased electricity was not directly linearly related to any one parameter. It was linked to temperature through a quadratic equation obtained through non-linear regression as described by Sagar [17].

The temperature/month was seen to have a U shaped relationship to building energy demand. Closer inspection revealed two concentrated superimposed curves. Therefore, the influence of combinations each of the different time parameters was checked by splitting the data into a series of 8 test cases as described in Table 4. Day time was defined as 5–19 h and night time was defined as 1–4, and 20–24 h. This was based on the parameter sunlight over the course of a year, as values higher than 0 were recorded from hours 5–19. Similar graphs were prepared as during data exploration using the ggpairs package.

**Table 4.** Details of eight cases of data splitting to examine the effects on purchased electricity.

| Case # | Day of Week | Time of Day | Holiday |
|--------|-------------|-------------|------------|
| 1 | Weekday | Day time | No holiday |
| 2 | Weekday | Day time | Holiday |
| 3 | Weekday | Night time | No holiday |
| 4 | Weekday | Night time | Holiday |
| 5 | Weekend | Day time | No holiday |
| 6 | Weekend | Day time | Holiday |
| 7 | Weekend | Night time | No holiday |
| 8 | Weekend | Night time | Holiday |

*2.8. Electricity Prices*

To calculate the price per hour to correspond to specific time periods, a formula in Excel was used (Appendix C). This formula initially checks if it is either a holiday or Sunday, subsequently if it is not summer and which time of day it is (two corresponding price categories), and finally if it is summer and which time of day it is (three corresponding price categories). The monthly electricity prices per time category (summer daytime, summer peak time, day time, and holiday/night time, all including the Feed in Tariff surcharges) were stored in a separate sheet, as were the known holidays.

Note that special care should be taken when applying the known electricity prices per hour during the days surrounding maintenance in November for two reasons: the total electricity use is uncharacteristically low, and the micro-grid is also disconnected during at least one day, which means that solar power generation is missing during that day. This leads to abnormal total prices for both parameters if these values are not excluded from the datasets.

To check the validity of the found relationships for the two micro-grid parameters, the average of the two equations found in previous steps were entered into the actual data to calculate hypothetical values for both solar power generation and purchased electricity, which were consequently linked to electricity price. Given the uncertainty of the improvement of SVR, the linear regression equation was applied to solar power generation. Due to the negative intercept (a) negative values for solar power generation were changed to 0. The results were compared with the actual data from 2015 to 2017, and their differences described.

## 3. Results

### 3.1. Data Cleansing (See SM5: R Code for Data Rows Graph)

Both 2015 and 2017 had 365 (days)*24 (hours) = 8,760 rows of parameters for hourly data. However, 2016 was a leap year and therefore included February 29, which led to 366 (days)*24 (hours) = 8,784 rows of parameters. After data cleansing the amount of data remaining from 2015, 2016, and 2017 was 98.54%, 98.36%, and 97.93% respectively. (See Figure 3, based on code from [18]).

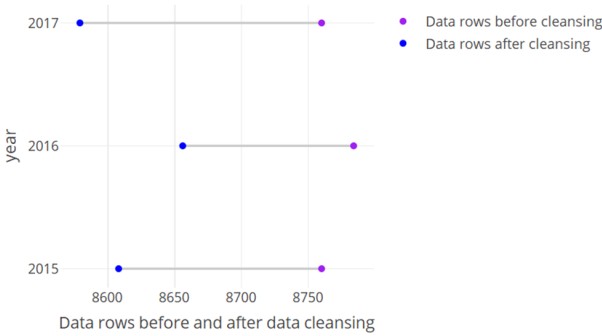

**Figure 3.** Amount of data before and after cleansing.

The differences in the standard deviation (SD), mean, minimum and maximum values of five parameters before outlier removal and after the final removal of uncharacteristic values are shown in Table 5.

**Table 5.** Standard deviation, mean, minimum and maximum values of five parameters before outlier removal and after the final data cleansing step.

| Parameter | | 2015 | | 2016 | | 2017 | |
|---|---|---|---|---|---|---|---|
| | | Pre-Cleansing | Post-Cleansing | Pre-Cleansing | Post-Cleansing | Pre-Cleansing | Post-Cleansing |
| Wind speed | SD | 1.304764 | 1.188131 | 1.266143 | 1.167836 | 1.267896 | 1.131123 |
| | Mean | 2.285958 | 2.233074 | 2.265922 | 2.226386 | 2.235808 | 2.174749 |
| | Min | 0 | 0 | 0 | 0 | 0 | 0 |
| | Max | 10 | 6.9 | 10.5 | 6.9 | 11.1 | 6.6 |
| Temperature | SD | 8.567907 | 8.599083 | 8.737433 | 8.760915 | 9.043145 | 9.069151 |
| | Mean | 14.9803 | 14.98214 | 14.95232 | 14.96025 | 14.28081 | 14.32673 |
| | Min | −6.3 | −6.3 | −6.0 | −6.0 | −6.5 | −6.5 |
| | Max | 35.6 | 35.6 | 35.5 | 35.5 | 34.4 | 34.4 |
| Solar irradiation | SD | 0.9289794 | 0.9274603 | 0.9168617 | 0.9157871 | 0.9353226 | 0.9356301 |
| | Mean | 1.058775 | 1.049586 | 1.058818 | 1.056141 | 1.103574 | 1.094547 |
| | Min | 0.01 | 0.01 | 0.01 | 0.01 | 0.01 | 0.01 |
| | Max | 3.54 | 3.54 | 3.64 | 3.64 | 3.63 | 3.63 |
| Total pv ac | SD | 18.35157 | 18.27055 | 17.2445 | 17.20114 | 17.07312 | 17.04483 |
| | Mean | 23.1429 | 22.89964 | 22.08021 | 22.02548 | 22.62292 | 22.40334 |
| | Min | 0.000466667 | 0.000466667 | 0.000466667 | 0.000466667 | 0.000466667 | 0.000466667 |
| | Max | 66.8908 | 66.8908 | 66.43907 | 66.43907 | 63.05009 | 63.05009 |
| Purchased electricity | SD | 111.1321 | 109.3086 | 122.9744 | 120.7636 | 115.5117 | 113.9781 |
| | Mean | 597.4223 | 598.4304 | 648.3106 | 649.815 | 692.5064 | 693.0694 |
| | Min | 212.8308 | 377.3867 | 218.2507 | 409.1167 | 276.9287 | 459.414 |
| | Max | 1138.182 | 1045.367 | 1105.39 | 1105.39 | 1133.443 | 1133.443 |

These results show that removing outliers of wind speed, total pv ac and purchased electricity does not greatly affect the standard deviation of all five parameters. It is interesting to note that whereas the maximum of solar irradiation has increased from 2015 to 2016 and stayed nearly the same between 2016 and 2017, the maximum solar power generated (total pv ac) is decreasing every year.

## 3.2. Data Exploration

The top of Figure 4 shows the amount of generated solar power per month and hour of the day, with values color coded by temperature; the bottom shows the amount of building energy demand for the same criteria; both for the year 2015. The top shows the expected relation of more solar power being generated during hours of sunlight. The bottom shows a clear baseline of energy demand related to storage and other research related equipment in the building, with peaks in energy demand in winter and summer months for the heating, ventilation, and air-conditioning system.

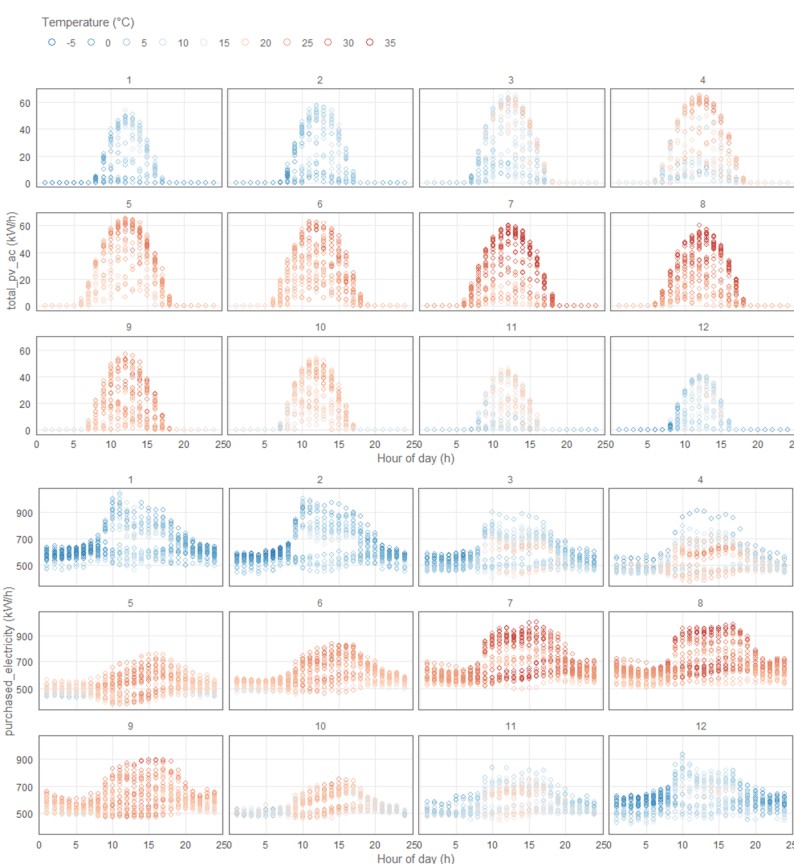

**Figure 4.** Generated solar power/building energy demand per month and hour of the day, with values color coded by temperature; 2015 data.

Figure 5 shows a cross comparison between 13 tested factors for correlation, with the data from 2015. For all years, solar power generation and solar irradiance were found to be highly correlated to each other (0.988–0.991), and both were found to be highly correlated to sunlight, and inversely correlated to humidity. They were both weakly correlated to temperature and wind speed. Other weather factors were also correlated in various degrees to each other and to the time parameters month and hour. Purchased electricity (building energy demand without micro-grid contributions) was found to be highly correlated to the day of the week, hour of the day, and temperature extremes (high from July to September, and especially in the morning hours of December to February). It is weakly correlated to holidays and other weather parameters sunlight, wind speed, solar irradiation, and humidity (inversed).

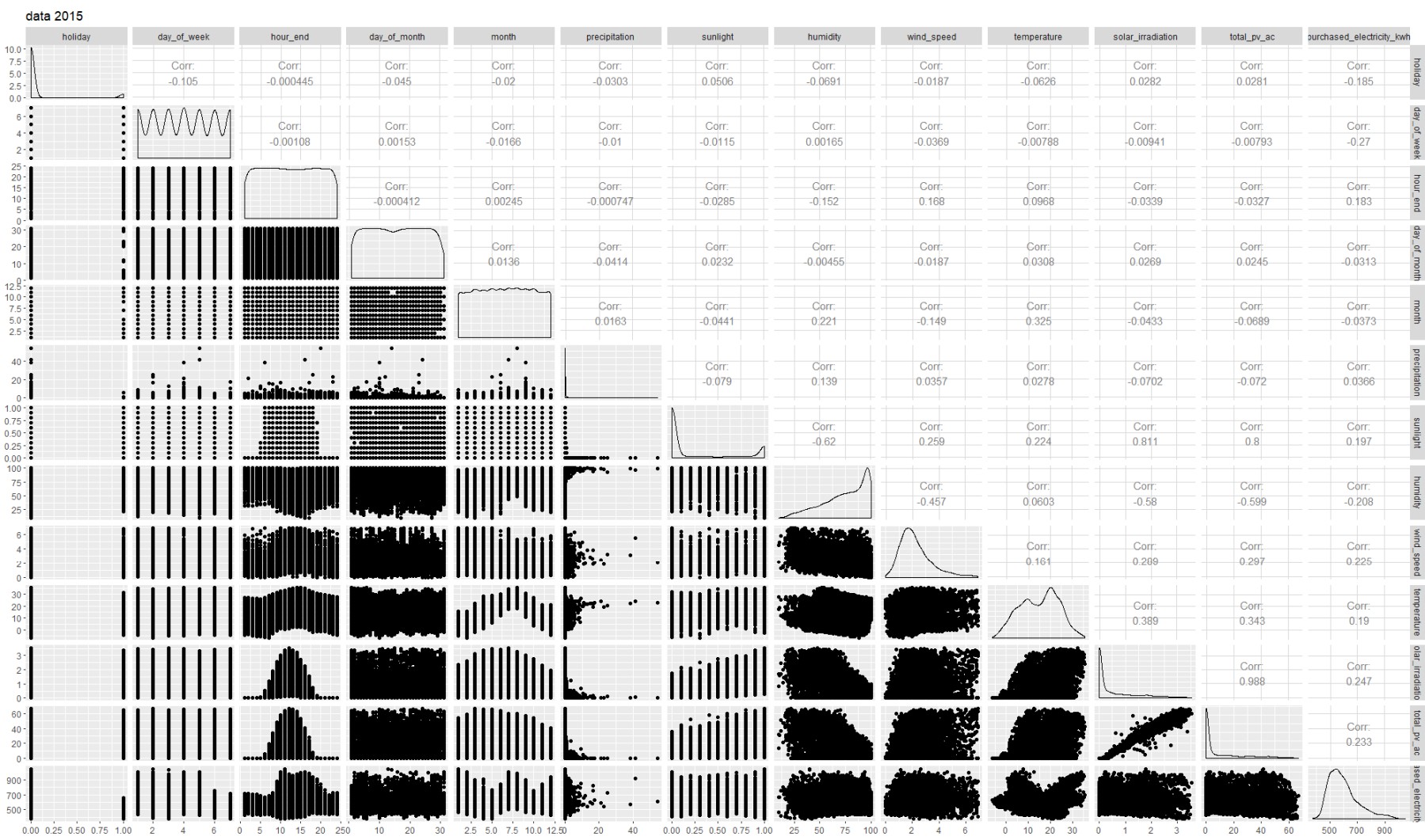

**Figure 5.** All parameters correlated to each other through the ggpairs package; 2015 data.

### 3.3. Solar Power Generation

The linear model had the format of (y = a + b * x). Smoothing the curve through means of support vector regression (red line, formed by x symbols, in Figure 6) shows a better fit than through the linear model (LM) (blue line, formed by x symbols, in Figure 6). Table 6 shows the difference in Root Square Mean Errors (RSME) for each year, proving the worth of applying support vector regression for 2015 and 2016, though it takes some computing time. For 2017, surprisingly the linear regression model showed less error than the SVR. This means that SVR cannot be applied haphazardly but must be tested against linear regression models. The values for epsilon and cost in Table 6 and Figure 7 show the epsilon parameters were all low, leaning towards overfitting of the models. The cost parameter found in 2016 was rather high, which also creates allows for few errors during classification. As SVR generates new models randomly, it may be necessary to compare the results of several runs repetitively even though one execution already creates dozens of models, in order to examine the differences.

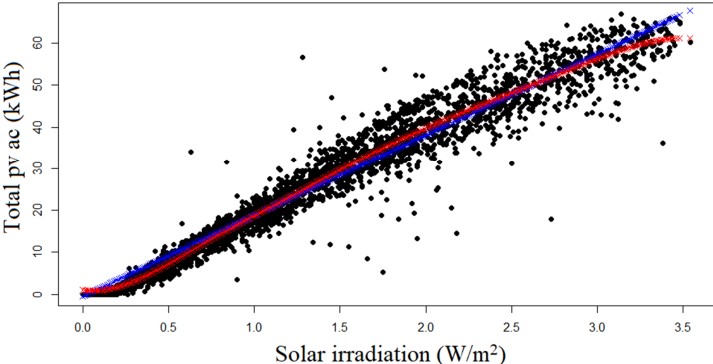

**Figure 6.** Linear (blue line) and Support Vector (red line) Regression of two parameters for 2015 data.

**Table 6.** Coefficients and Root Square Mean Errors (RSME) and parameters of Linear Models (LM) (y = a + b * x), Support Vector Regression (SVR), and tuned SVR models.

| Year | Intercept (a) | Slope (b) | LM RSME | SVR RSME | Tuned SVR RSME | Epsilon | Cost |
|------|---------------|-----------|---------|----------|----------------|---------|------|
| 2015 | −0.5101 | 19.2985 | 2.527038 | 2.37958 | 2.263242 | 0.08 | 8 |
| 2016 | −0.5088 | 18.4918 | 2.204646 | 1.974913 | 1.888035 | 0.06 | 128 |
| 2017 | −0.4922 | 18.0549 | 2.181584 | 2.37958 | 2.263037 | 0.08 | 4 |
| Average | −0.5037 | 18.61506667 | | | | | |

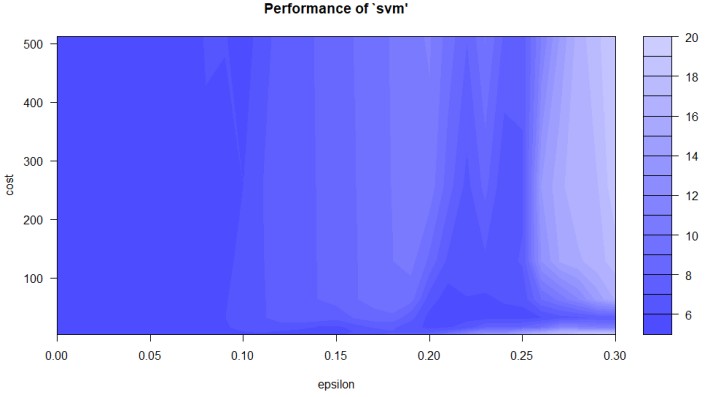

**Figure 7.** Epsilon and costs factors by SVR; 2015 data.

### 3.4. Building Energy Demand

Figure 8 shows the equation found through non-linear regression to best describe the relationship between purchased energy and temperature for 2015. This equation has the format

of $y = a - b * x + c * x^2$, where y is purchased electricity, x is temperature, a is the intercept, b is the slope, and c the exponential curve. Results from all years are shown in Table 7, including the error difference between linear and non-linear models. The non-linear model error is lower than the linear model error for all cases. However, the data seem to show two superimposed u-shaped curves, suggesting a third parameter is affecting the results and it may be possible to generate two equations to better model the data. To enable this, the data were separated into 8 cases as described in Table 4. The subsequent results showed no clear effect of any combination of parameters on the purchased electricity. Figure 9 shows the effect of temperature, month, and hour on the purchased electricity from 2015. This illustrates there is no distinguishable repetitive difference in effects between temperature and hours of the day during which the building is occupied on the purchased electricity, which was expected.

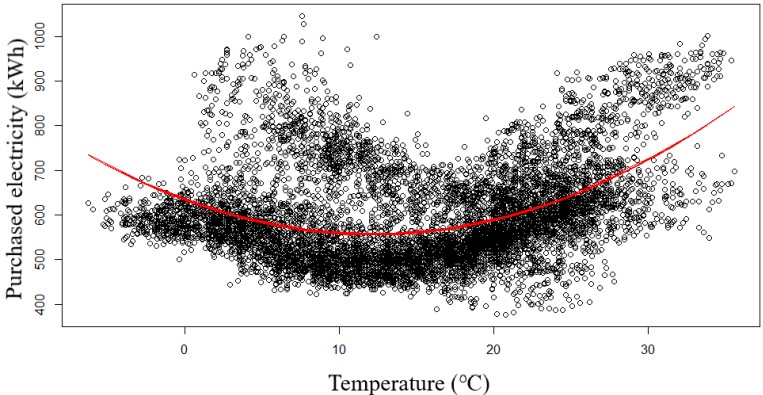

**Figure 8.** Quadratic equation relating purchased electricity to temperature; 2015 data.

**Table 7.** Parameters for the quadratic equation (6) describing the relationship between temperature and purchased electricity. All had 1 iteration to convergence.

| Year | LM Error | NLM Error | Intercept (a) | Slope (b) | Curve (c) | Residual Sum of Squares | Tolerance |
|------|----------|-----------|---------------|-----------|-----------|-------------------------|-----------|
| 2015 | 107.3204 | 99.04455 | 633.1747 | 12.7784 | 0.5251 | 84,442,961 | $1.83 \times 10^{-9}$ |
| 2016 | 113.8353 | 103.4177 | 660.1028 | 13.3542 | 0.6305 | 92,577,853 | $4.283 \times 10^{-9}$ |
| 2017 | 109.8714 | 102.973 | 698.3077 | 9.8411 | 0.4722 | 90,956,240 | $8.543 \times 10^{-9}$ |
| Average | | | 663.8617333 | 11.99123333 | 0.5426 | | |

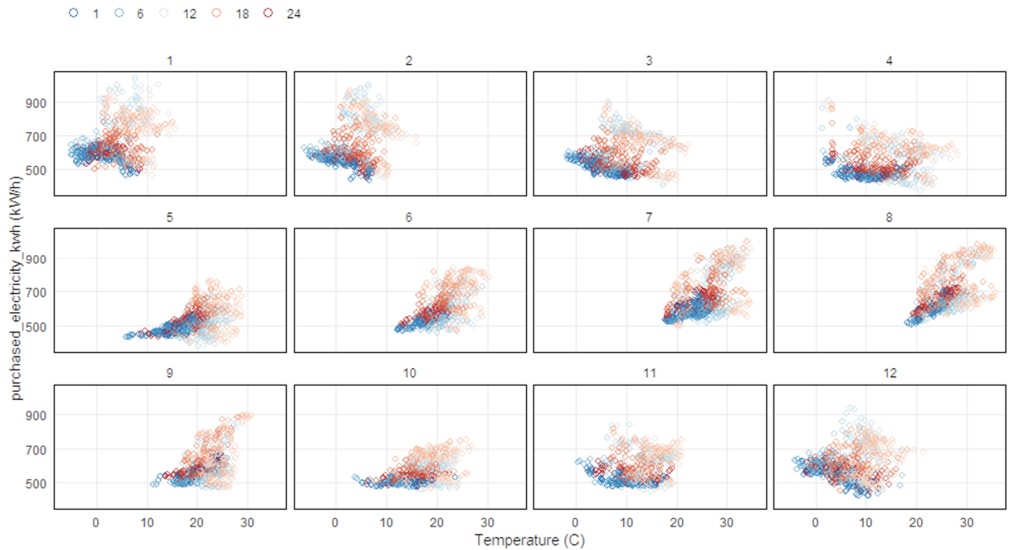

**Figure 9.** Purchased electricity per month, temperature, and hour of the day; 2015 data.

### 3.5. Electricity Prices

Table 8 shows the results of applying the predicted equations for solar power generation and purchased electricity to the electricity price totals in comparison to the actual electricity price totals. The equations' accuracy varied from +6 to −8% of actual prices, with purchased energy showing the greatest variation.

**Table 8.** Electricity prices in 1000 JPY and % correctly modeled per micro-grid parameter (costs saved vs. remaining costs) for both actual data and estimated data based on modeled equations.

| Parameter | 2015 | 2015 | % | 2016 | 2016 | % | 2017 | 2017 | % |
|---|---|---|---|---|---|---|---|---|---|
| total_pv_ac (costs saved) | 1579 | 1557 | −1.36% | 1594 | 1643 | +3.08% | 1651 | 1740 | +5.41% |
| purchased_electricity_kwh (costs made) | 86,669 | 92,049 | +6.21% | 99,253 | 97,304 | −1.96% | 107,646 | 99,142 | −7.90% |

## 4. Discussion

Two different types of models were used to estimate the relationships between the micro-grid parameters on the one hand, and weather and time parameters on the other. This is due to the dependencies found in the exploratory data analysis phase.

This analysis showed a linear relationship between solar power generation and solar irradiance, which was not always improved by applying support vector regression. The effects of temperature on solar power generation were different than anticipated. Figure 4 shows that whereas higher temperatures of 30–35 degrees Celsius indeed lead to less produced solar power (a maximum of around 60 kWh whereas colder temperatures showed higher production); lower temperatures of −5-0 degrees Celsius lead to far less produced solar power of 40 kWh. Although the efficiency may be higher, the lack of solar irradiation has a far greater effect than temperature. The electricity costs saved by generated solar power were overestimated by 3% in 2016 and 5% in 2017, suggesting that degradation has a considerable effect on the total generated solar power and, although hidden, should be taken into account in future models.

To examine the hypothesis that temperature could have a large influence on purchased energy, a quadratic equation was found to describe this relationship. At times when the building was occupied, temperature extremes did show a higher effect on purchased electricity (Figure 4). Applying the equation to hourly electricity costs led to over- and underestimations of the actual costs of 6% and 8% respectively. With annual costs running up to around 100,000,000 JPY, it is desirable to have an accuracy within 1% or less, which this simplified equation has not yet achieved. Purchased energy was found to be dependent on numerous other variables, namely, month, hour of day, day of week, holiday or not, and other weather parameters. Splitting the data into groups as per table IV did not lead to a clear distinction, as the month of the year still had a substantial influence as shown in Figure 9. A more inclusive approach incorporating all of these parameters might produce better results, especially if the hidden factor of building occupancy, or of purchased electricity base demand, is estimated for each hour.

To enable predictions of up to one year into the future by means of machine learning, the first step in improving these equations are to incorporate solar power generation efficiency degradation on the one hand, and the complex interactions of time parameters leading to building occupancy on the other hand. If these improvements can lead to more accurate cost predictions within 1% error margin, the second step is to predict associated weather parameters for up to one year, mainly solar irradiance and temperature. As these two parameters influence each other, and this study showed that purchased electricity had a higher correlation to solar irradiance than to temperature (Figure 5), it may even be possible to have a single weather parameter predict both micro-grid parameters. This would simplify the number of unknowns that have to be predicted. Alternatively, if many weather factors can be predicted with great accuracy, the reverse approach may be taken. Sharma et al. [5] found that solar irradiance can be inferred from the parameters day of the year, temperature, dew point,

wind speed, sky cover (alternatively sun hours), precipitation, and humidity. If the input weather parameters can be predicted with sufficient accuracy, it is worthwhile to compare the results to using only solar irradiance as input.

Although linear, non-linear, and support vector regression were adequate in uncovering the relationships and the strength of relationship between the various parameters, it was not yet successful in defining the complex interplay that leads to purchased electricity. A future study applying principal component regression, such as done by Davo et al. [19], may resolve how to best model purchased electricity with the greatest accuracy combined with the least number of parameters.

## 5. Conclusions

The equations resulting from linear, non-linear, and support vector regression allowed for long-term (one year) predictions of related electricity costs that were within 7.9% accuracy of actual costs. This study further found evidence supporting the potential to model both solar power generation and building energy consumption simultaneously, based on the parameter of solar irradiance and in combination with known time parameters affecting building occupancy.

This study is the first to attempt to predict both energy demand and production on a long-term scale using R software in combination with actual micro-grid data records. The cross correlations generated between all input parameters and generated equations provide important indications as to how to integrate energy demand and production into one model with potentially one weather parameter, solar irradiance, as input. For future studies on building energy consumption and solar power generation, this study provides an important basis for evaluating which weather and time parameters to consider during data exploration, as well as correlation methods. It furthermore shows how applying actual data records can assist directly in improving model accuracy. Finally, all the documentation on R codes used during the data exploration and analysis phases is provided as appendices and Supplementary Materials, which facilitates replicating the steps taken with other datasets.

**Supplementary Materials:** The following are available online at http://www.mdpi.com/2076-3417/10/13/4462/s1, SM1: R code for outlier removal, SM2: R code for data exploration graphs, SM3: R code for linear regression and SVR, SM4: R code for plotting nonlinear regression; SM5: R code for data rows graph. Datasets related to this article can be found at https://doi.org/10.6084/m9.figshare.c.4176698, hosted at figshare (2018), or https://doi.org/10.11503/nims.1003, hosted at NIMS IMEJI (2018).

**Author Contributions:** Conceptualization, K.V. and Y.K.; methodology, K.V.; software, K.V.; validation, K.V.; formal analysis, K.V. and E.A.; investigation, K.V. and E.A.; data curation, E.A. and K.V.; writing—original draft preparation, K.V.; writing—review and editing, K.V., E.A., and Y.K.; visualization, K.V.; supervision, Y.K.; project administration, K.V. All authors have read and agreed to the published version of the manuscript.

**Funding:** Part of the research is financially supported by the MEXT Program for Integrated Materials Development.

**Conflicts of Interest:** The authors declare no conflict of interest.

**Notes on References:** In order to encourage gender equality in STEM research, the authors have included a minimum of 50% references of which the first author appears to be female. For more information about this goal, see: Collins, K. Conversation with Dr. Emily M. Gray. Journal and Proceedings of the Gender Awareness in Language Education Special Interest Group 2019, 11, pp. 71-80, SSN 1884-152X. Feb. 2019. URL: https://gale-sig.org/wp-content/uploads/2019/04/Winter-18-19-Full-Issue-v4.pdf.

## Appendix A

**Table A1.** Differing forecasting horizons from previous prediction studies.

| Parameter | Long Term | Medium Term | Short Term | Reference |
|-----------|-----------|-------------|------------|-----------|
| Solar irradiance | 1–3 days | 1–6 h | 15 min–2 h | [20] |
| Solar irradiance | 28–45 h (1 day) | 1–6 h | - | [21] |
| Solar irradiance | 1 month | 1 day | 1 min | [22] |
| Building energy demand | 1–10 years | 1 month–1 year | 1 h–1 week | [23] |
| Building energy demand | > 1 year | 1 week–1 year | 1 h–1 week | [24] |

## Appendix B

**Table A2.** Advantages and disadvantages of different software for supervised machine learning [25,26].

| Software | R | Python | Java | SAS | SPSS |
|---|---|---|---|---|---|
| Description | Developed by statisticians, aimed at handling numerical values; procedural language (step by step subroutines) | General purpose programming language, simple to read, used for websites; object oriented (bundles procedures and data together as parts of objects) | Programming language and computer platform; object oriented (bundles procedures and data together as parts of objects) | Statistical Analysis System, developed for analyzing large quantities of agriculture data | Statistical Package for the Social Sciences, the first statistical programming language for the PC |
| Advantages | Free, easy to learn, many options for data visualization, suited for statistical analysis and explanatory and predictive modeling | Free, easy to learn, built-in debugging | High speed | Easy to handle large datasets | Good user interface, official support, easy to learn and write code, similar to Excel |
| Disadvantages | No official support or user interface | Slow with large datasets | Not strong in statistical modeling and visualization | High learning curve | Slow with large datasets |

## Appendix C

Excel formula for electricity prices:
IF(OR(K3 = 1,I3 = "Sunday"),VLOOKUP(C3,electricityprice!C$2:H$74,5,FALSE),
IF(AND(J3<>"Summer",P3 > = 8,P3 < = 21),VLOOKUP(C3,electricityprice!C$2:H$74,4,FALSE),
IF(AND(J3<>"Summer",P3<=7),VLOOKUP(C3,electricityprice!C$2:H$74,5,FALSE),
IF(AND(J3<>"Summer",P3>=22),VLOOKUP(C3,electricityprice!C$2:H$74,5,FALSE),
IF(AND(J3 = "Summer",P3 > = 13,P3 < = 15),VLOOKUP(C3,electricityprice!C$2:H$74,3,FALSE),
IF(AND(J3 = "Summer",P3 > = 16,P3 < = 21),VLOOKUP(C3,electricityprice!C$2:H$74,2,FALSE),
IF(AND(J3 = "Summer",P3 > = 8,P3 < = 12),VLOOKUP(C3,electricityprice!C$2:H$74,2,FALSE),
IF(AND(J3 = "Summer",P3 < = 7),VLOOKUP(C3,electricityprice!C$2:H$744,5,FALSE),
IF(AND(J3 = "Summer",P3 > = 22),VLOOKUP(C3,electricityprice!C$2:H$74,5,FALSE),
-999999)))))))))

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
