# Peer review of "Long-Term Forecasting Potential of Photo-Voltaic Electricity Generation and Demand Using R"

_applsci, doi:10.3390/app10134462_

Round 1

Reviewer 1 Report

The aim of this paper is to determine the potential of long-term forecasting (at least one year) of energy production and demand on the scale of a research building by means of programming with the software R and applying linear, non-linear, and support vector regression.

The paper is written in a clear way, very easy to understand. The data and ideas proposed are novel and fit new applications that can be made using Machine Learning. The study of different software tools for the use of machine learning is very clear. The references are appropriate, and they are up-to-date.

The resolution of figures 4, 5 and 9 could be improved. It would be better to put a legend in axis x and y in figures 6, an 8.

Reviewer 2 Report

The paper describes how the authors used statistical models to relate the photo-voltaic electricity generation and the demand of a building to environmental variables (temperature, solar irradiation and others). The statistical analysis is the key part of the study and involves preprocessing the data and fitting linear regression and support vector regression models. It is positive that the authors seem to make an effort to make the data analysis reproducible and transparent. Unfortunately, I did not receive the R code of the statistical analysis and I did not see a link to a public git repository. Thus, I cannot judge whether the statistical analysis is solid and reproducible. Moreover, the study does not fully exploit the statistical tool regarding model and variable selection. And the statistics and machine learning literature is not cited properly. Therefore, I recommend that the statistical analysis and its presentation is improved before the article is published in this journal.

Detailed comments:
- I recommend having the R code checked by an expert before the article is published. Only fully reproducible studies should be published. Yhe R code was not available to for review.
- Title: with 'R programming' I associate the development of new software. Maybe 'Long-term Forecasting Potential of Photo-voltaic Electricity Generation and Demand using R' or 'Long-term Forecasting Potential of Photo-voltaic Electricity Generation and Demand using the statistical software R' is better.
- A statistical method for variable and model selection is missing. Which variable do significantly improve the model? Is the nonlinear relationship improving the performance of the model significantly?
- A comparison of statistical software is out of the scope of the paper and all content related to that should be removed. I recommend that the author simply write that they are using R and add a proper reference for R including the version number.
- "The electricity costs saved by generated solar power were overestimated by 3% in 2016 and 5% in 2017, suggesting that degradation has a considerable effect on the total generated solar power and, although hidden, should be taken into account in future models." --> This could be due to changes of the weather and electricity price and possibly other factors. How do you separate those effects?
- I recommend adding solid reference to statistical publications and books instead of web tutorials. E.g. instead of citing [15] (a link to an online tutorial) you could used doi: http://dx.doi.org/10.18637/jss.v015.i09 It might be worth to think about teaming up with a statistician to improve the data analysis and its presentation.
- line 197: "or amount of points allowed within the epsilon margin" is wrong.
- Fig. 4 is of bad quality (every thing seems to be blurred).
- Fig. 5: It is difficult to read the labels.
- Fig. 6: what is 'y' and what is 'x'? --> Improve labels. And the lines are rather 'x' symbols than lines.
- Fig. 6: The fit of the SVR looks non-linear. However, a description of the non-linearity is missing in Section 2.6.
- Fig. 8: what is 'y' and what is 'x'? --> Improve labels.
- Fig. 9: Text in gray can be difficult to read.

Reviewer 3 Report

There are several points that need further clarification and improvement. Although, the language and presentation of the results are in general satisfactory, there are several small points in the paper that need editorial improvement. My main comments are summarized as follows:

1) More items should be added to clear the comparison in Table B1.

2) Figures 4 and 5 need to change and more explanations should be provided.

3) Some technical solutions should be provided to discuss how to deal with the comparison between the demand and the production energy.

4) Some recent papers on adaptive protection must be considered:

[A] M. K. Jena, S. R. Samantaray and B. K. Panigrahi, "A New Adaptive Dependability-Security Approach to Enhance Wide Area Back-Up Protection of Transmission System," in IEEE Transactions on Smart Grid, vol. 9, no. 6, pp. 6378-6386, Nov. 2018, doi: 10.1109/TSG.2017.2710134.

[B] H. F. Habib, N. Fawzy, M. M. Esfahani and O. A. Mohammed, "Enhancement of Protection Scheme for Distribution System Using the Communication Network," 2019 IEEE Industry Applications Society Annual Meeting, Baltimore, MD, USA, 2019, pp. 1-7, doi: 10.1109/IAS.2019.8911872.

[C] R. Jain, D. L. Lubkeman and S. M. Lukic, "Dynamic Adaptive Protection for Distribution Systems in Grid-Connected and Islanded Modes," in IEEE Transactions on Power Delivery, vol. 34, no. 1, pp. 281-289, Feb. 2019, doi: 10.1109/TPWRD.2018.2884705.

[D] H. F. Habib, N. Fawzy and O. A. Mohammed, "A Fault Clearing for Microgrid Protection System Utilized the Communication Network with Centralized Approach," 2019 SoutheastCon, Huntsville, AL, USA, 2019, pp. 1-4, doi: 10.1109/SoutheastCon42311.2019.9020644.

Since, a lot of work has been done on this topic, the authors must emphasize and discuss the distinct advantages of the proposed protection scheme against others in the literature.

Round 2

Reviewer 2 Report

I already gave comments in the first round of the review and I acknowledge that the authors have made efforts to improve the manuscript. If the editors are ok with publishing a study that bases its statistical analyses on non peer-reviewed online tutorial instead of the adequate statistical literature, the manuscript is ready for publication. Otherwise it will be necessary to revise the statistical analysis with a statistical expert. Since I am not familiar with the exact requirements of the journal, I give the overall recommendation 'accept in present form' below. However, I would also understand it when the editors decide to condition the acceptance one a revision of the statistical analysis.